# The NADPH Oxidase A of *Verticillium dahliae* Is Essential for Pathogenicity, Normal Development, and Stress Tolerance, and It Interacts with Yap1 to Regulate Redox Homeostasis

**DOI:** 10.3390/jof7090740

**Published:** 2021-09-09

**Authors:** Vasileios Vangalis, Ioannis A. Papaioannou, Emmanouil A. Markakis, Michael Knop, Milton A. Typas

**Affiliations:** 1Department of Genetics & Biotechnology, Faculty of Biology, National and Kapodistrian University of Athens, 15784 Athens, Greece; vasvagg@biol.uoa.gr; 2Center for Molecular Biology of Heidelberg University (ZMBH), University of Heidelberg, 69120 Heidelberg, Germany; m.knop@zmbh.uni-heidelberg.de; 3Laboratory of Mycology, Department of Viticulture, Vegetable Crops, Floriculture and Plant Protection, Institute of Olive Tree, Subtropical Crops and Viticulture, N.AG.RE.F., Hellenic Agricultural Organization-DIMITRA, 71307 Heraklion, Crete, Greece; markakis@elgo.iosv.gr; 4German Cancer Research Center (DKFZ), DKFZ-ZMBH Alliance, 69120 Heidelberg, Germany

**Keywords:** antioxidant response, conidial anastomosis tubes (CATs), heterokaryon incompatibility, oxidative stress, pathogenicity, reactive oxygen species (ROS), redox homeostasis

## Abstract

Maintenance of redox homeostasis is vital for aerobic organisms and particularly relevant to plant pathogens. A balance is required between their endogenous ROS production, which is important for their development and pathogenicity, and host-derived oxidative stress. Endogenous ROS in fungi are generated by membrane-bound NADPH oxidase (NOX) complexes and the mitochondrial respiratory chain, while transcription factor Yap1 is a major regulator of the antioxidant response. Here, we investigated the roles of NoxA and Yap1 in fundamental biological processes of the important plant pathogen *Verticillium dahliae*. Deletion of *noxA* impaired growth and morphogenesis, compromised formation of hyphopodia, diminished penetration ability and pathogenicity, increased sensitivity against antifungal agents, and dysregulated expression of antioxidant genes. On the other hand, deletion of *yap1* resulted in defects in conidial and microsclerotia formation, increased sensitivity against oxidative stress, and down-regulated antioxidant genes. Localized accumulation of ROS was observed before conidial fusion and during the heterokaryon incompatibility reaction upon nonself fusion. The frequency of inviable fusions was not affected by the deletion of Yap1. Analysis of a double knockout mutant revealed an epistatic relationship between *noxA* and *yap1*. Our results collectively reveal instrumental roles of NoxA and ROS homeostasis in the biology of *V. dahliae*.

## 1. Introduction

Reactive oxygen species (ROS) are short-lived, highly reactive molecules that are produced by partial reduction of oxygen, and they include hydrogen peroxide (H_2_O_2_), superoxide anion (O_2_^•−^) and hydroxyl (OH^•^) radicals [1]. Growing evidence from various organisms has assigned a Janus-faced nature to ROS as both deleterious molecules that can cause irreversible damage to biological systems and important signaling components regulating metabolic and developmental processes [1,2,3]. This double-edged significance of ROS is clearly illustrated in host–microbe interactions. In particular, fungal plant pathogens have to cope with host-derived ROS burst reactions during infection [4], while endogenous ROS production is simultaneously involved in the development of infectious structures and the efficient colonization of their hosts [5]. It is, therefore, vital for fungal pathogens to maintain a balance between ROS generation and scavenging in order to achieve successful development and pathogenicity [5,6].

Intracellular ROS production mainly takes place in mitochondria, as by-products of the electron transport chain [7], and by the enzymatic activity of NADPH oxidase (Nox) complexes [1]. The latter mostly localize at the plasma membrane or at the endoplasmic reticulum, and they produce superoxide anion radicals via reduction of molecular oxygen, using NADPH as the electron donor [1]. The best characterized member of this family is the mammalian gp91^phox^ (Nox2), which is responsible for the ROS burst of neutrophils and other phagocytic cells in response to microbial pathogens [8]. Filamentous fungi possess three well established families of Nox enzymes called NoxA, NoxB (homologs of the mammalian gp91^phox^), and NoxC. These enzymes, together with their adapter protein NoxD, form distinct complexes with pleiotropic roles in sexual reproduction, vegetative growth, and host infection [5,6,9,10,11,12,13]. In particular, NoxA has been implicated in sexual differentiation of fruiting bodies in *Aspergillus nidulans* [14], *Podospora anserina* [15], and *Sordaria macrospora* [16], while its deletion in *Neurospora crassa* has led to complete female fertility [17]. Generation of ROS by NoxA is indispensable for appressorium-mediated cuticle penetration in *Magnaporthe oryzae* [10], as it controls the polarized elongation of penetrations pegs [11]. Similarly, a complete lack or high attenuation of pathogenicity was observed upon deletion of *noxA* in *Claviceps purpurea* [18], *Sclerotinia sclerotiorum* [19], *Botrytis cinerea* [20], and *Alternaria alternata* [21]. Finally, NoxA has been proposed to be involved in the control of somatic conidial fusion via conidial anastomosis tubes (CATs) in various species [22], including the plant pathogens *Verticillium dahliae* [23], *Fusarium oxysporum* [24], and *B. cinerea* [25].

In parallel with ROS-generating systems, fungi have evolved a fine-tuned arsenal to cope with oxidative stress. Basic leucine zipper (bzip) transcription factors are conserved across eukaryotic life and are implicated, among others, in the oxidative stress response (OSR), developmental processes, amino acid biosynthesis, and nutrient utilization [26,27]. The AP1 family is the largest and best characterized group of fungal bzip transcription factors [28], and it includes Yap1, the major regulator of the OSR, which orchestrates several ROS-scavenging enzymes and non-enzymatic antioxidants [5]. Extensive studies of *Saccharomyces cerevisiae* Yap1 revealed that oxidation of specific cysteine residues upon exposure to H_2_O_2_ is crucial for the accumulation of the protein in the nucleus, providing a characteristic example of ROS-mediated signal transduction [29]. Homologs of Yap1 in filamentous fungi have similar structures and their deletion increases sensitivity to oxidative agents, indicating the conserved role of Yap1 in the regulation of the OSR [28]. Transcriptional analyses in several species have shown that Yap1 homologs control the expression of antioxidant genes such as catalases, superoxide dismutases, peroxidases, and genes involved in glutathione biosynthesis [30,31,32,33,34,35,36,37]. Apart from their universal role in regulating the OSR, Yap1 homologs also exhibit species-specific roles in pathogenicity [33,36,38,39], development [33,37,39,40], and secondary metabolism [35,41].

The asexual plant-pathogenic fungus *V. dahliae* causes wilt disease in a wide range of economically important plants [42]. Upon induction from a plant host, its resting structures (i.e., microsclerotia) in the soil germinate and form infectious hyphae that attach to a host’s surface and penetrate its roots by forming swollen hyphae (termed hyphopodia) and penetration pegs [42,43]. The formation of these pegs depends on a ROS burst derived from the NoxB/Pls1 membrane-bound complex [44]. During host colonization, *V. dahliae* must cope with the plant-derived oxidative stress that is induced by plant defense mechanisms [45], which renders efficient OSRs necessary for successful systemic infection. However, a recent study of *V. dahliae* Yap1 showed that its deletion did not impair the ability of this fungus to cause disease in smoke trees [40].

In this work, we aimed at the functional characterization of *noxA* and its potential involvement in important biological processes of *V. dahliae*. We coupled this investigation with further analyses of *yap1*, which has recently been studied in *V. dahliae* [40], to gain insight into the possible interplay between ROS-generating and -scavenging systems in fungi. To this end, we knocked out their *V. dahliae* homologs and further generated a double knockout mutant to investigate their roles in fungal development, physiology, and pathogenicity. Furthermore, we characterized the responses of these mutants to antifungal agents and their behavior during hyphal fusion and the heterokaryon incompatibility reaction. Our results attribute significant roles to NoxA and Yap1 regarding morphogenesis, pathogenicity, and stress tolerance, and they highlight the crucial contribution of endogenous ROS production and metabolism to multiple aspects of fungal biology.

## 2. Materials and Methods

### 2.1. Fungal Strains, Growth Media, and Culture Conditions

All fungal strains constructed and used in this study are listed in Appendix A. They were grown on standard growth media (Potato Dextrose Agar—PDA, Czapek-Dox complete medium—CM, Czapek-Dox minimal medium—MM) at 24 °C, in the dark. Preparation and maintenance of monoconidial strains have been described previously [46].

### 2.2. Deletion and Complementation of V. dahliae noxA and yap1

All plasmids used for the construction of recombinant vectors are listed in Appendix A. We have recently described the strategy that we used for knocking out the *V. dahliae* homologs of *noxA* and *yap1* [23]. Briefly, both mutants were constructed in the background of *V. dahliae* wild-type strain 123V via double homologous recombination following *Agrobacterium tumefaciens*-mediated transformation (ATMT) [47]. Using this method, we recently performed and described deletion of *noxA* and validation of the mutant strain [23]. To knock out the *yap1* homolog, the flanking chromosomal regions (~2.0 kb-long each) of the *yap1* open reading frame (ORF) were amplified from genomic DNA of *V. dahliae* 123V and ligated to the *neo*^R^ cassette (conferring resistance to geneticin; the cassette was amplified from plasmid pSD1 [48]) in the backbone of the *A. tumefaciens* binary vector pOSCAR [49]. All PCR amplification steps were performed using the high-fidelity Herculase II Fusion DNA Polymerase (Agilent, Santa Clara, CA, USA), and assembly of the recombinant vector was performed with the NEBuilder HiFi DNA Assembly Master Mix (New England Biolabs, Ipswich, MA, USA). To generate the double Δ*noxA* Δ*yap1* knockout mutant, we transformed the Δ*noxA* strain with the deletion construct of *yap1*. Validation of all knockout strains was achieved with PCR, using gene-specific primers (Appendix A), and Southern hybridization analyses (DIG DNA Labeling and Detection Kit, Sigma-Aldrich, St. Louis, MO, USA).

The coding sequence of the *yap1* gene, flanked by ~2.0 kb-long genomic regions, was amplified from genomic DNA of *V. dahliae* 123V and co-transformed into protoplasts of the Δ*yap1* strain with plasmid pUCATPH [50] (which carries the *hph* cassette conferring resistance to hygromycin B) to generate the complemented strain *yap1*-c.

### 2.3. Characterization of Morphology, Physiology, and Stress Response of V. dahliae Strains

Morphological and physiological characterization of fungal strains, as well as assessment of their stress sensitivity, were performed as previously described [47], with a minor modification for the determination of germination frequency. In particular, strains were grown for 12 h in CM and then checked microscopically for the emergence of germ tubes.

To characterize stress tolerance and responses, we exposed fungal strains to a variety of oxidative agents (H_2_O_2_, paraquat, iprodione, and *N*-acetyl cysteine), substances that induce osmotic stress (NaCl and sorbitol), cell wall damaging factors (amphotericin B obtained from Biosera, Nuaille, France, fluconazole from Pfizer, Brooklyn, NY, USA, calcofluor white M2R, and Congo red), trace elements (CaCl_2_, CuSO_4_, and FeSO_4_), and an inhibitor of TOR kinase (sirolimus from Cayman Chemical, Ann Arbor, MI, USA). In addition, we used fungicides that target complex I (sodium amytal), complex II (isopyrazam, flutolanil), complex III (azoxystrobin, kresoxim-methyl, pyraclostrobin-QoI/cytB site; annisulborn-Qi site), and complex IV (sodium cyanide) of the electron transport chain to investigate possible interactions between the ROS-generating systems. All chemicals were purchased from Sigma-Aldrich, St. Louis, MO, USA, unless otherwise specified above. According to our previously described methods [47], we used spot dilution assays and determined relative growth inhibition (i.e., % growth inhibition = ((colony diameter on CM − colony diameter in stress condition)/(colony diameter on CM)) × 100). All experiments were performed at least in triplicate.

### 2.4. Virulence Assays

Plant pathogenicity bioassays were performed according to our previously published methods [47,51]. Briefly, eggplant seedlings were inoculated with the corresponding fungal strain by soil drenching (20 mL of conidial suspension of 5.0 × 10^6^ conidia/mL per pot). Plants were maintained at 24 °C with a 12 h light/dark cycle. Assessment of disease severity at each time point (up to 40 days) and determination of relative AUDPC and plant growth parameters were performed according to previously described protocols [51,52]. We re-isolated fungal strains from the treated plants by transferring xylem chips (three chips from nine randomly selected plants per treatment) to acidified PDA plates. The isolation ratio was expressed as the number of xylem chips from each treatment that exhibited fungal growth.

### 2.5. Cellophane Penetration Assays

The ability of *V. dahliae* strains to penetrate cellophane membranes was assessed as follows. Conidial suspensions from the examined strains were collected from 7-day-old PDA cultures, and 5 × 10^6^ conidia of each were transferred onto CM plates overlaid with sterile cellophane sheets. The samples were incubated at 24 °C for five days. Cellophane sheets were then removed, and the plates were incubated for four additional days before being scored for fungal growth. For the microscopic examination of hyphopodial formation, the same membranes (i.e., after five days of incubation with the respective conidial suspension) were washed with sterile water and observed under a microscope.

### 2.6. Superoxide Detection

Superoxide anion radicals (O_2_^•−^) in germlings and in the mycelium of *V. dahliae* strains were detected by a nitro blue tetrazolium chloride (NBT; Cayman Chemical, Ann Arbor, MI, USA) staining assay. Conidia of each strain were grown for 16 h in CM, in plate wells with sterile coverslips at their bottom. Following incubation, the medium was removed, 5 mL of a 0.2% NBT solution was added, and the samples were incubated at room temperature (RT), in the dark, for 45 min. The coverslips were then washed with ethanol and checked microscopically. Detection of superoxide anion radicals in the mycelium was performed with the addition of 10 mL of a 0.2% NBT solution on 20-day-old cultures grown on CM. Plates were incubated for 45 min at RT, which was followed by washing with ethanol and incubation for another 30 min, before being air-dried and scored. Staining assays were performed in duplicate.

### 2.7. Reverse Transcription Quantitative PCR (RT-qPCR)

Determination of the expression levels of selected genes was performed as follows. Wild-type, Δ*noxA*, Δ*yap1*, and Δ*noxA* Δ*yap1* strains were grown for five days in CM at 24 °C. Mycelia were collected, washed with water, and when desired treated with 1.5 mM H_2_O_2_ for 45 min. They were then snap-frozen in liquid nitrogen (N_2_), ground to a fine powder under N_2_, and used for extraction of total RNA with the NucleoSpin RNA kit (Macherey-Nagel, Düren, Germany). Total RNA was reverse transcribed to cDNA using the PrimeScript cDNA synthesis kit (Takara Bio, Kusatsu, Japan). Quantitative PCR (qPCR) was performed using the KAPA SYBR Fast Universal Master Mix (Roche, Basel, Switzerland) in a Mx3000P real-time PCR instrument (Stratagene California, San Diego, CA, USA), according to the manufacturer’s instructions. For each primer pair (Appendix A), we performed optimization of oligo concentrations and construction of standard curves in preliminary experiments. All curves were highly linear (R^2^ > 0.999), and their amplification efficiencies ranged between 95% and 105%. The cycling protocol consisted of an initial denaturation step at 95 °C for 5 min, followed by 40 cycles of 10 s at 95 °C (denaturation), 20 s at 60 °C (annealing), and 20 s at 72 °C (elongation). Dissociation curves were generated for each primer pair with the following protocol. Samples were first incubated for 1 min at 95 °C, then at 55 °C for 30 s, and finally a temperature ramp (0.1 °C/s) up to 95 °C was applied with continuous collection of fluorescence readings. A single product was amplified with each primer pair. No-template control samples were included in each run to check for contamination and significant formation of primer dimers.

The 2^−^^ΔΔCt^ method [53] was used for analysis of the results, with the modification that each ΔCt value was calculated as the difference between the Ct values of the reference and target genes (i.e., Ct reference-Ct target), and the fold change was calculated by the formula 2^ΔΔCt^. The *V. dahliae* β-tubulin gene (VDAG_10074) was used as the internal reference in all experiments. The ΔΔCt value and fold change for each gene in different mutants and/or conditions was calculated relative to the control condition (i.e., expression level of the corresponding gene in the untreated wild-type strain). Student’s *t*-tests were used to assess differences of ΔΔCt values. Three biological (i.e., independent cultures) and two technical replicates were performed for each strain and condition.

### 2.8. Quantification of CAT-Mediated Fusion

We used our previously optimized and described methods for reproducible quantification of CAT-mediated self and nonself fusion [23,54] to investigate the possible involvement of *noxA* and *yap1* in somatic cell fusion. Briefly, conidia from 7-day-old PDA cultures were collected and transferred to 6-mm Petri dishes, with coverslips at their bottom (containing 5 mL of CAT medium each) to a final concentration of 2.0 × 10^6^ conidia/mL. Plates were incubated for 60 h at 24 °C before imaging. Each strain/pairing was tested in triplicate and 200 fusions were recorded per replicate.

### 2.9. Microscopy

Microscopic examination of fungal germlings, hyphae, and NBT-stained samples, as well as investigation of CAT-mediated fusion, were performed using a Zeiss Axioplan epifluorescence microscope equipped with a differential interference contrast (DIC) optical system, a set of filters BP450-490 (excitation) and BP515-595 (emission), and a Zeiss Axiocam MRc5 digital camera. Methylene blue staining (0.005% *w/v*) was used to differentiate live from dead cells; samples were incubated at 25 °C for 5 min (in the dark) before imaging.

## 3. Results

### 3.1. Roles of NoxA and Yap1 in V. dahliae Morphogenesis and Physiology

The NADPH oxidase A (NoxA) and the transcriptional regulator Yap1 are key components of ROS metabolism in fungi and other organisms. The single homolog of *noxA* in the *V. dahliae* genome (VDAG_06812 in the reference genome of strain Ls.17) codes for a predicted protein of 555 aa with a high similarity to its *N. crassa* Nox1 homolog (99% query coverage, 90% sequence similarity), and we have recently described its deletion in the wild-type *V. dahliae* strain 123V [23]. The *A. nidulans* Yap1 protein sequence was used as a query in tBlastN genomic searches for the identification of its single *V. dahliae* homolog, which encodes a predicted protein of 583 aa (VDAG_01588; protein similarity: 52%). This protein is predicted to contain the characteristic bzip domain and the signal peptides of its *S. cerevisiae* homolog, as well as the conserved cysteine residues that are essential for its nuclear localization (Appendix A). For the functional analysis of *yap1* in *V. dahliae*, the gene was deleted from strain 123V, and the resulting deletion mutant (Δ*yap1*) was validated by PCR and Southern blot analyses (Appendix A). The wild-type *yap1* gene was re-introduced into Δ*yap1* to generate the complemented strain *yap1*-c. A double deletion Δ*noxA* Δ*yap1* mutant was constructed via double homologous recombination following transformation of Δ*noxA* conidia with the *yap1* deletion construct (Appendix A).

Morphogenesis and physiology of strains Δ*noxA*, Δ*yap1*, and Δ*noxA* Δ*yap1* were compared to those of their wild type on growth media PDA, CM, and MM. Production of microsclerotia was significantly reduced in mutants Δ*noxA* and Δ*noxA* Δ*yap1* on PDA, while it was almost absent from Δ*yap1* on all media (Figure 1A). The ability of Δ*noxA* and Δ*noxA* Δ*yap1* to produce aerial hyphae was compromised on CM and MM (Figure 1A), and these strains also exhibited moderately slower growth on PDA (*p* < 0.01 and *p* < 0.05, respectively; Figure 1B). In contrast, deletion of *yap1* did not affect the formation of aerial mycelium, but it slightly reduced its growth rate (*p* < 0.05, Figure 1A,B).

Regarding development of conidia, a non-significant increase in their production was observed in Δ*noxA*, but their ability to germinate was drastically limited in this mutant (51.3% reduction), similarly to what was observed in Δ*noxA* Δ*yap1* (43.7% reduction) (Figure 1B). On the other hand, Δ*yap1* exhibited a 10-fold reduction in conidiation (Figure 1B), although its conidia germinated normally (Figure 1B,C). We also observed that mature hyphae of Δ*noxA* and Δ*noxA* Δ*yap1* were significantly thinner than those of the wild type (Figure 1D,E). All these defects were fully rescued in the corresponding complemented strains (Figure 1A,B,E).

### 3.2. NoxA, but Not Yap1, Is Essential for the Penetration Ability and Pathogenicity of V. dahliae

To investigate the possible involvement of NoxA and Yap1 in *V. dahliae* pathogenicity, we assessed the ability of the deletion mutants to cause disease in eggplant, in comparison to their wild type. Both Δ*noxA* and Δ*noxA* Δ*yap1* mutants exhibited diminished pathogenicity, with 88% and 76% of the treated plants (respectively) remaining completely asymptomatic 40 days after inoculation, while the remaining plants showed only weak symptoms (Figure 2A,B). In contrast, deletion of *yap1* only slightly limited the fungal potential to cause disease (Figure 2A,B). Complementation of the corresponding mutants with the wild-type alleles of *noxA* and *yap1* fully rescued the ability of the fungus to cause severe disease, characterized by defoliation, wilting, and chlorosis (Figure 2A,B).

We further performed a time-course analysis of virulence to gain a better understanding of the involvement of *noxA* and *yap1* in the induction of plant disease (Figure 2C). The wild-type strain started causing observable symptoms in the infected plants 12 days after inoculation, reaching a mean disease severity of 55.5% (±7.3%) at the end of the experiment (29 days), and an overall relative AUDPC value of 27.5% (±3.9%) (Figure 2C,D). Deletion of *noxA* completely prevented the expression of disease symptoms during this time period, similarly to the Δ*noxA* Δ*yap1* strain, which achieved a relative AUDPC score of 0.1% (±0.1%) (Figure 2C,D). In contrast, only a minor reduction was detected in virulence of Δ*yap1* (Figure 2C,D). These findings were further supported by the determination of the average plant fresh weight at the end of the experiment, with Δ*noxA* and Δ*noxA* Δ*yap1* having no effect, whereas Δ*yap1* led to a reduction similar to that of the wild-type strain (Figure 2E). Consistently, our attempts to re-isolate the fungus from xylem chips (29 days post-inoculation) revealed a significantly reduced presence of the pathogen in the xylem vessels of plants inoculated with Δ*noxA* or Δ*noxA* Δ*yap1*, in contrast to Δ*yap1*, which achieved systemic colonization at wild-type levels (Figure 2F).

The inability of the Δ*noxA* and Δ*noxA* Δ*yap1* strains to cause disease in their plant hosts, as well as their severely reduced presence in the xylem of inoculated plants, led us to the hypothesis that *noxA* is possibly necessary for efficient root penetration. We therefore investigated the capacity of the deletion mutants to penetrate cellophane membranes by inoculating cellophane sheets overlaying CM plates with conidia of each strain and incubating them for 5 days. Cellophane sheets were then removed, plates were incubated for another 4 days, and they were finally scored for fungal growth. In support of our hypothesis, deletion of *noxA*, or both *noxA* and *yap1*, completely abolished penetration of cellophane, in contrast to the wild-type, the Δ*yap1*, and the complemented strains, which grew profoundly on but also below the membrane sheets (Figure 3A). Furthermore, microscopic examination of the removed membranes revealed a drastic reduction in the frequency of normally developed hyphopodia (i.e., infectious structures involved in host penetration [43]) in Δ*noxA* and Δ*noxA* Δ*yap1*, whereas all other strains retained the ability to form numerous hyphopodia (Figure 3B). Our results indicate that NoxA, but not Yap1, plays a major role in plant penetration and has essential functions in *V. dahliae* pathogenicity.

### 3.3. NoxA Is Involved in ROS Metabolism and Its Oxidative Activity Is Counteracted by Yap1

Staining of O_2_^•−^ radicals using NBT revealed their accumulation mostly in the apical parts of germlings and at the periphery of colonies of the Δ*noxA* mutant, in contrast to their more uniform presence in the wild-type strain (Figure 4A). Peripheral staining of colonies was also observed in Δ*noxA* Δ*yap1*, whereas the wild-type pattern was restored in the complemented strain (Figure 4A). These observations support the hypothesis that NoxA is involved in ROS metabolism in *V. dahliae*. On the other hand, deletion of *yap1* resulted in generally increased formazan precipitation (Figure 4A), which indicates defects in ROS metabolism and is consistent with the conserved role of this regulator in ROS detoxification.

We further characterized the roles of NoxA and Yap1 in response to oxidative stress by exposing the wild-type and knockout strains to the oxidizing agent H_2_O_2_, the herbicide paraquat, the fungicide iprodione, and the ROS scavenger N-acetyl cysteine (NAC). As expected, we detected significant growth defects of Δ*yap1* in the presence of any ROS-inducing agent (Figure 4B,C). On the other hand, Δ*noxA* was inhibited to a lesser extent by H_2_O_2_ (at high concentrations), iprodione, and the ROS-scavenger NAC (Figure 4B,C). Notably, the double deletion mutant Δ*noxA* Δ*yap1* exhibited the same behavior as Δ*noxA* (Figure 4B,C), indicating an epistatic relationship between the two genes.

To gain a better understanding of the roles of *V. dahliae noxA* and *yap1* in the OSR, we investigated their expression patterns before and after treatment with H_2_O_2_ by RT-qPCR (Figure 5). The *noxA* gene is strongly induced upon oxidative stress in the wild type, and neither its induction pattern nor its transcript levels were affected by deletion of *yap1*. On the other hand, *yap1*, which is down-regulated under oxidative stress in the wild type, failed to reach its normal expression levels in the Δ*noxA* mutant background under these conditions. When we also analyzed the expression levels of the stress-activated Hog1 MAP kinase gene before treatment with H_2_O_2_, we observed significantly lower levels in all knockout mutants than in the wild type. Under oxidative stress, however, *hog1* was down-regulated in the wild type, followed a similar but less pronounced trend in Δ*yap1*, and moderate induction in Δ*noxA* and Δ*noxA* Δ*yap1* (Figure 5).

Furthermore, we characterized the responses to oxidative stress of three important downstream genes known to be involved in the OSR, i.e., catalase (*cat1*), superoxide dismutase (*sod1*), and glutathione reductase (*glr1*). As expected, induction of all three was detected in the wild-type strain in the presence of H_2_O_2_, whereas they were expressed at lower levels and failed to be induced by oxidative stress in the absence of their transcriptional regulator Yap1 (Figure 5). Interestingly, we detected significantly higher mRNA levels of *cat1* and *glr1* in the Δ*noxA* mutant than in the wild type under normal conditions, while *sod1* and *glr1* failed to reach their wild-type levels under oxidative stress in Δ*noxA* (Figure 5). The Δ*noxA* Δ*yap1* strain exhibited a similar expression pattern to Δ*yap1* for *cat1* and *glr1*, while the expression of *sod1* was up-regulated under normal conditions in this strain (Figure 5). Our findings indicate that *V. dahliae* NoxA is up-regulated by oxidative stress and implicate it in the transcriptional regulation of both the OSR and the important stress-responsive MAP kinase Hog1.

### 3.4. NoxA Is Involved in the Response to Cell Wall Stress and Possibly Interacts with Components of Important Biological Processes

We further investigated the possible involvement of NoxA and Yap1 in fungal responses to other sources of cellular stress by characterizing the behavior of the corresponding mutant strains in the presence of substances inducing hyperosmotic stress, cell wall-perturbating agents, high concentrations of trace metals, and various fungicides (Figure 6A,B). Regarding osmotic stress, only a minor growth inhibition of Δ*noxA* and Δ*yap1* was recorded in the presence of NaCl. In contrast, cell wall stress significantly compromised the growth rates (calcofluor white and Congo red) and the germination frequency (calcofluor) of Δ*noxA* and Δ*noxA* Δ*yap1*, whereas Δ*yap1* exhibited limited sensitivity only to calcofluor. Consistently, the drug amphotericin B, which disorganizes fungal cell membranes by targeting ergosterol, and also causes oxidative stress, impaired growth and germination of Δ*noxA* and Δ*noxA* Δ*yap1*. However, the inhibitor of ergosterol biosynthesis fluconazole specifically restricted growth of Δ*yap1* and, to a lesser extent, that of the double deletion mutant. Motivated by the observation that deletion of *yap1* in yeast causes decreased resistance to metals [55], we checked resistance of the *V. dahliae noxA* and *yap1* mutants to Ca^2+^, Cu^2+^, and Fe^2+^ cations (Appendix A). While Ca^2+^ and Cu^2+^ indeed restricted radial growth of Δ*yap1*, Fe^2+^ had a more pronounced effect on the *noxA-*deficient strains.

Apart from the Nox complexes, mitochondria are a major intracellular source of ROS. We tested fungicides that target the four complexes (I–IV) of the respiratory chain (i.e., azoxystrobin, sodium cyanide, pyraclostrobin, kresoxim-methyl, annisulborn, sodium amytal, flutolanil, and isopyrazam), and we found them all to cause significant growth reduction in Δ*noxA* and Δ*noxA* Δ*yap1* strains, while Δ*yap1* was not affected (Figure 6A,B; Appendix A). These findings are suggestive of a possible crosstalk between the two major cellular sources of ROS.

Finally, since NoxA has been shown to interact with the machinery of autophagy in phagocytic cells [56], we tested the responses of the *V. dahliae* mutants to rapamycin, an inhibitor of the TORC1 kinase and, therefore, an inducer of autophagy (Figure 6A,B). Deletion of *noxA*, but not of *yap1*, led to significantly increased sensitivity to the inhibitor, which implies a possible dysregulation of autophagy.

### 3.5. NoxA, but Not Yap1, Is Essential in Both Partners for CAT-Mediated Cell Fusion

We have previously found that NoxA in *V. dahliae* is essential for somatic fusion of conidia or germlings via CATs [23]. On the contrary, deletion of *yap1* exerted no effect on CAT-mediated fusion (Figure 7A). The double deletion mutant exhibited the same behavior as Δ*noxA* (Figure 7A). Surprisingly, the Δ*noxA* mutant retained its capacity for hyphal fusion between mature hyphae (Figure 7B), which suggests differences between the regulatory mechanisms of conidial and hyphal fusion.

We hypothesized that the role of NoxA in cell fusion via CATs could be related to the generation or perception of the unknown signal that is involved in pre-fusion cell communication. If that process were unidirectional (i.e., each fusion partner emitting or receiving the signal, but not both), a functional copy of *noxA* in only one of the interacting partners could be sufficient for fusion of a subset of cells. To test this possibility, we paired the deletion strains with the strain Ls.17 H1-sgfp (expressing GFP-tagged histone H1 in its nuclei), which permitted the microscopic identification of nonself fusions. The presence of functional *noxA* genes in both partners was found to be essential for their fusion (Figure 7C); deletion of *yap1* from one partner showed no effect, as expected.

It is unclear whether NoxA functions in cell fusion by causing localized ROS production with direct effects on partner recognition or the commitment to fuse, or indirectly by transducing signals to alter the expression of other genes or the activity of their products. We found that, during the homing phase of the interacting cells prior to their fusion, O_2_^•−^ always accumulated at the tips of the mutually attracted CATs (Figure 7D), suggesting that localized generation of ROS putatively by NoxA is indeed involved in the process. No ROS accumulation was detected after completion of CAT-mediated fusion (Figure 7D). On the other hand, deletion of *noxA* had no effect on the expression levels of the MAP kinases Fus3 and Slt2, which are important components of the pre-fusion communication mechanism in *N. crassa* [22], and are also required for CAT fusion in *V. dahliae* [23].

Fusion of vegetatively incompatible strains of fungi, including *V. dahliae*, usually triggers an incompatibility reaction that can cause cell death of the anastomosed compartments [54]. Based on the observations that this reaction in *N. crassa* is characterized by an induction of ROS [57], and that genes involved in the OSR are up-regulated during CAT formation in *Colletotrichum gloeosporioides* [58], we used NBT staining to study O_2_^•−^ accumulation in incompatible *V. dahliae* fusions (pairing Ls.17 H1-sgfp × 123V). One-third of such fusions (n = 30) exhibited increased formazan precipitation (Figure 7F), indicating that ROS accumulation is indeed associated with the onset of the incompatibility reaction, similarly to what was observed in *N. crassa* [57]. Deletion of *yap1* had no effect on the frequency of inviable fusions (Ls.17 H1-sgfp × 123V Δ*yap1* pairing), which suggests either that the observed ROS accumulation does not mediate the catastrophic reaction or that Yap1 does not respond to it.

## 4. Discussion

The dual nature of ROS in biological systems as a potential cause of cellular damage on the one hand, and as signaling components of important developmental processes on the other, has been the focus of extensive research [1,2,5,8]. This duality is clearly illustrated in the case of fungal plant pathogens, which need to cope with their host-derived ROS bursts during infection [4], while they generate and use endogenous ROS for their own development and pathogenicity [5,6]. Therefore, the survival and ecological success of these organisms depend on their ability to maintain a redox balance that presumably requires coordination of ROS generation and scavenging systems. In this study, we investigated the roles of the ROS-producing NADPH oxidase NoxA in the biology of the important plant-pathogenic fungus *V. dahliae*, in comparison to its transcription factor Yap1, a conserved central regulator of ROS detoxification [13,28]. We found that NoxA has multifaceted roles in important developmental and physiological processes, plays an essential role in virulence, mediates responses to different types of cellular stress, and is required for somatic cell fusion.

Fungal NADPH oxidase complexes have been attributed pleiotropic functions in morphogenesis and pathogenicity [9,12,13,59]. In *V. dahliae*, we found NoxA to be required for normal formation of microsclerotia, i.e., dormant resting structures that are crucial for long-term survival and spread of the species [42]. In addition, *V. dahliae* NoxA is necessary for normal formation of aerial hyphae and germination of its asexual spores, in agreement with similar previously described roles in, e.g., *N. crassa*, *Epichloë festucae*, *C. purpurea*, and *S. macrospora* [16,17,18,60]. Furthermore, our investigation revealed that NoxA is essential for virulence in *V. dahliae*, and our findings suggest that this can be at least partly due to its involvement in the formation of hyphopodia, i.e., infectious swollen hyphae that can penetrate the roots of the plant hosts to initiate the disease cycle [43,44]. Although the regulatory mechanisms that underlie the development of *V. dahliae* hyphopodia are mostly unknown, recent reports implicated in this process both Ste11, a component of MAP kinase signaling, and Csin1, a cellophane surface-induced protein that acts through the cAMP pathway [61,62]. These findings suggest putative interactions of ROS with important signaling transduction pathways. Previous studies have demonstrated the involvement of NoxA in the formation of appressoria in the plant pathogen *M. oryzae* [10,11] and traps in the nematode pathogen *Arthrobotrys oligospora* [63]. These findings together indicate conserved involvement of the ROS-generating enzyme NoxA in the early stages of infection of diverse pathogenic fungi.

Another explanation for the highly compromised ability of *V. dahliae* to cause systemic infection and disease symptoms in the absence of NoxA could be its increased sensitivity to various types of environmental stress, most notably cell wall and oxidative stress. Oxidative bursts of ROS have been described to play a role in cell wall biosynthesis [9], which could be linked to the sensitivity of the *V. dahliae* Δ*noxA* mutant to cell wall-perturbating agents. Our analyses also revealed reduced resistance of Δ*noxA* to oxidative stress, which could be attributed to dysregulation of antioxidant genes, such as the example of *sod1* that we observed in our study. In addition, deletion of *noxA* resulted in dysregulation of the important stress-activated MAP kinase Hog1, which could be directly or indirectly linked to some of the identified pleiotropic phenotypes of NoxA. These observations could be likely attributed to the expected function of ROS as secondary signals that could potentially control the expression and function of multiple transcription factors and signaling components via post-translational modification [2]. A possible mechanism for this could involve the control of cellular localization of key signaling components, such as the MAP kinase Mpk1 in *P. anserina*, whose normal localization depends on *nox1* [64]. Notably, the absence of functional NoxA rendered the fungus sensitive to several antifungal agents, including representatives that block the central TOR kinase, as well as the four complexes of the mitochondrial electron transport chain. The latter observation indicates that the maintenance of a minimum level of intracellular ROS is important for cell functionality and viability. It could also be relevant to the crosstalk that has been demonstrated between mitochondrial ROS and NADPH oxidases in mammalian cells [65], hypothetically reflecting the function of mechanisms that ensure the coordination of those two major intracellular sources of ROS in the context of oxidative homeostasis.

Conidial and hyphal fusion is an integral component of the establishment and development of fungal colonies [66], and nonself fusion could possibly grant access to parasexual generation of diversity [23,54]. We have previously demonstrated that NoxA is essential for CAT-mediated fusion of *V. dahliae* conidia or germlings [23], and we further found in the present study that the requirement for functional NoxA applies to both interacting cells. We also detected O_2_^•−^ accumulation at the tips of wild-type germlings during their homing phase of interaction prior to fusion, which could most likely be attributed to the activity of NoxA. Similar observations have previously been made in *B. cinerea* [25]. We hypothesize that temporally and spatially regulated ROS bursts could be involved in the pre-fusion communication of the interacting cells, possibly linked to the oscillatory recruitment of signaling components (such as the MAP kinase Fus3 and the scaffold protein SOFT) to cell tips that controls cell fusion in *N. crassa* [67].

The transcription factor Yap1 has a conserved role in ROS detoxification among diverse organisms as a central regulatory element that participates in the orchestration of their OSR [30,31,32,33,34,35,36,37]. Indeed, we found deletion of *yap1* to result in increased sensitivity to all oxidative agents tested, as well as to the fungicide fluconazole, which can induce ROS production in *Cryptococcus neoformans* [68,69]. In addition, the expression of important antioxidant genes, such as *cat1*, *sod1*, and *glr1*, was lower and non-inducible by oxidative stress in the absence of Yap1. Apart from its regulatory function in oxidative stress tolerance, Yap1 has also been attributed additional developmental and other, sometimes contradictory, roles between fungi [28]. Here, we found that in *V. dahliae* Yap1 is necessary for normal conidiation, in line with previous findings in *M. oryzae* and *A. nidulans* [33,37], while its deletion also severely compromised the formation of microsclerotia. A previous study demonstrated attenuated microsclerotial formation upon *V. dahliae yap1* deletion [40], to a lesser extent than the one observed in our work, which probably reflects strain-specific variation in the fine-tuning of the underlying regulatory mechanisms. In agreement with the previous study that analyzed virulence on smoke trees [40], we detected no significant involvement of Yap1 in the virulence of *V. dahliae* on eggplant. This is consistent with studies of other fungal pathogens that do not require Yap1 for infecting their hosts (e.g., *B. cinerea*, *Fusarium graminearum*, and *Cochliobolus heterostrophus*), but in contrast to other fungi where Yap1 is indispensable for successful infection (e.g., *Ustilago maydis*, *M. oryzae*, and *Alternaria alternata*) [31,33,35,38,70,71]. These findings suggest that, in some fungi, alternative detoxification pathways or regulatory mechanisms likely take over during infection and suffice for efficient ROS detoxification when Yap1 is absent. Such alternatives could include the transcription factors Atf1 and Skn7, which mediate responses to RNS and ROS, respectively, by regulating the expression of antioxidant genes [72,73].

Based on the significant biological roles of the ROS-generating NADPH oxidase NoxA in *V. dahliae*, and the presumed need of the pathogen to withstand the ROS bursts involved in the defense mechanisms of its hosts [4], we would expect systems for coordination between ROS generation and processing to be important for its ROS homeostasis. In support of this hypothesis, our study revealed an antagonistic epistatic relationship between *noxA* and *yap1*, with their double null mutant exhibiting the same behavior as *ΔnoxA* in all tested phenotypes, instead of any detectable additivity between the two genes. This was also observed with regard to sensitivity to oxidative stress, whereby Δ*noxA* and Δ*noxA* Δ*yap1* were more resistant to oxidative agents than Δ*yap1*. Our interpretation of these findings is that Yap1 contributes to the neutralization of the ROS produced by NoxA, but this Yap1 detoxification function depends on NoxA, which itself responds to oxidative stress with higher expression levels. We propose that, in the absence of NoxA, other *yap1*-independent systems take over, presumably via up-regulation, to effectively shield the organism from the hazards of elevated oxidative stress. This is further supported by the detected changes in the expression patterns of antioxidant genes in the Δ*noxA* mutant. The observed small increase in Hog1 transcripts observed in this strain is unlikely to mediate antioxidant responses, since, in *V. dahliae*, Hog1 appears to play no role in the OSR [74]. In other fungi, oxidative stress signaling alternatives include Pap1 activation by Tpx1 and the Hog1 homologs Sty1/Spc1 MAPK pathway, which can trigger antioxidant responses in *Schizosaccharomyces pombe* [75]. Furthermore, the OSR of the filamentous fungus *A. nidulans* involves, in addition to its Yap1 homolog NapA, the Skn7 homolog SrrA and the Hog1 homologs SakA and MpkC [76,77,78]. Finally, we would attribute the findings that low levels of oxidative stress (e.g., low concentrations of H_2_O_2_) have no observable effect on the Δ*noxA* strain and, consistently, no induction of antioxidant genes, to the perturbated generation of endogenous ROS, which could be linked to a higher resistance of the fungus to low levels of exogenous oxidative stress.

In this study, we characterized the *V. dahliae* homolog of the important ROS producer NoxA, and we demonstrated significant pleiotropic roles in multiple developmental and physiological processes. We provide evidence that implicates NoxA in sensing and mediating responses to oxidative stress through genetic interactions with the transcriptional regulator Yap1 and presumably alternative, currently unknown, ROS signaling and detoxification systems. These data contribute to a better understanding of fungal ROS homeostasis and welcome future research to elucidate the significance and modes of action of ROS metabolism in the development, pathogenicity, and stress response of fungal pathogens.

## Figures and Tables

**Figure 1 jof-07-00740-f001:**
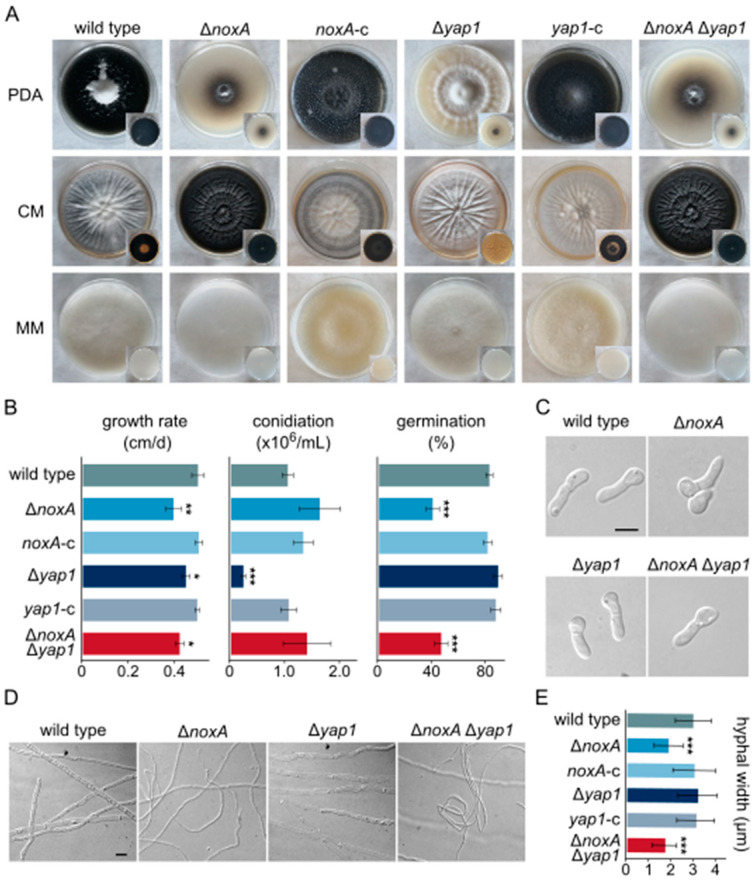
Morphological and physiological characterization of *V. dahliae* Δ*noxA*, Δ*yap1*, and Δ*noxA* Δ*yap1* deletion mutants: (**A**) morphology of colonies of the *V. dahliae* wild-type (123V), Δ*noxA*, Δ*yap1*, Δ*noxA* Δ*yap1,* and the complemented strains *noxA*-c and *yap1*-c after growth on PDA, CM, or MM for 35 days. (**B**) Growth rate, conidial production, and germination frequency of tested strains. All experiments were performed in triplicate, and germination of 100 conidia was assessed per replicate. Bars: SD. Statistical significance of differences from the wild type were tested with Student’s *t*-tests (* *p* ≤ 0.05, ** *p* ≤ 0.01, *** *p* ≤ 0.001). (**C**) Morphology of conidial germlings of the wild-type and the knockout strains. Bar = 5 μm. (**D**) Morphology of mature hyphae. Bar = 5 μm. (**E**) Mean hyphal width of examined strains. For each strain, the widths of at least 50 non-apical hyphal compartments from colony areas of the same age were recorded. Bars: SD. Statistical significance of differences from the wild type was assessed using Student’s *t*-tests (*** *p* ≤ 0.001).

**Figure 2 jof-07-00740-f002:**
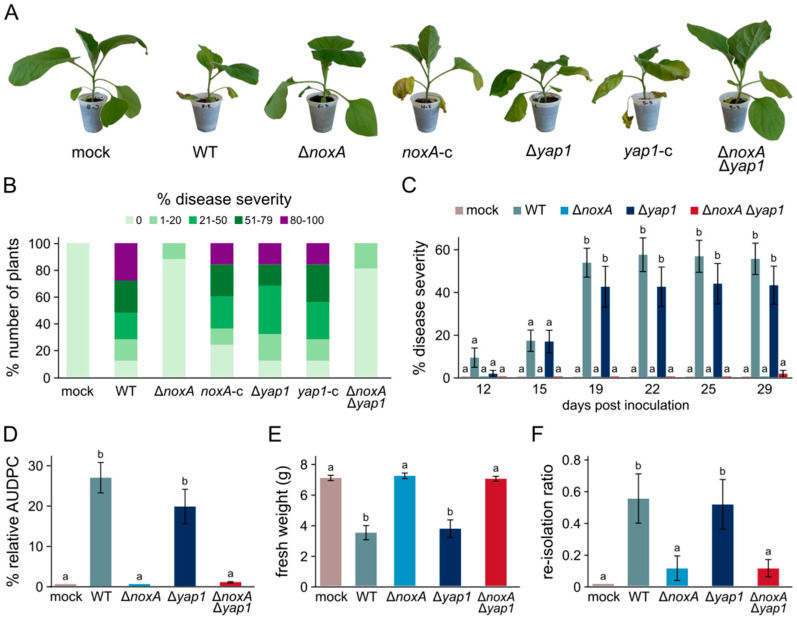
Phytopathological characterization of *V. dahliae noxA* and *yap1* knockout strains: (**A**) representative examples of inoculated plants with the wild-type strain 123V (WT), the deletion mutants Δ*noxA*, Δ*yap1*, and Δ*noxA* Δ*yap1*, as well as their corresponding complemented strains, 29 days post-inoculation. (**B**) Average disease severity caused by the examined strains, 40 days after inoculation (30 eggplant seedlings/strain). Non-infected plants (mock) served as controls. (**C**) Time-course analysis of disease severity caused by the wild-type and the knockout strains (21 plants/strain) over 29 days. (**D**) Mean relative area under disease progress curve (AUDPC) score of each strain (29 days). (**E**) Average plant fresh weight at the end of the time-course experiment (29 days). (**F**) Fungal re-isolation ratios at the end of the time-course experiment (29 days). Bars in (**C**–**F**): SE. Statistical significance of differences between strains was tested by one-way ANOVA followed by Tukey’s post hoc tests. Bars marked with the same letter did not differ significantly (*p* ≤ 0.05).

**Figure 3 jof-07-00740-f003:**
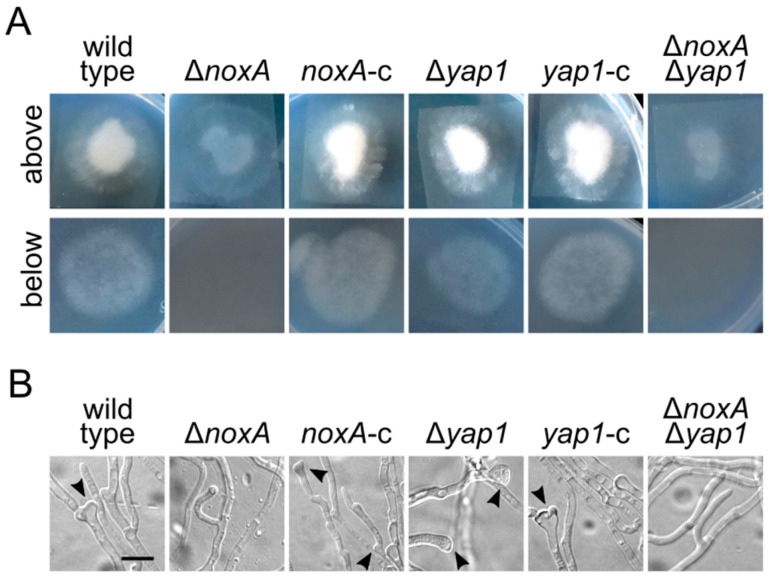
Penetration assays on cellophane membranes: (**A**) sheets of cellophane membrane were placed on CM agar, inoculated with conidial suspensions of the examined strains, and incubated for 5 days (above). Plates were scored for fungal growth 4 days after membrane removal (below). The experiments were performed in triplicate and 4 cellophane sheets per plate were used. (**B**) Microscopic examination of hyphopodial formation on the cellophane membranes (5 days post-inoculation). Arrowheads: examples of hyphopodia. Bar = 5 μm.

**Figure 4 jof-07-00740-f004:**
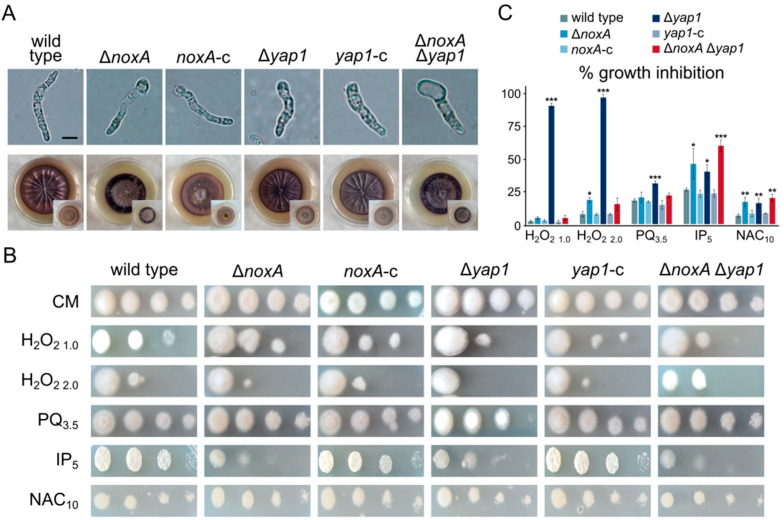
Patterns of ROS accumulation and oxidative stress tolerance of *V. dahliae noxA* and *yap1* knockout mutants: (**A**) detection of O_2_^•−^ using NBT staining in germlings (16 h of growth in CM) and colonies (20 days old on CM plates). Top and bottom views of plates are shown for each strain. (**B**) Conidial germination of each examined strain upon treatment with H_2_O_2_, paraquat, iprodione, or N-acetyl cysteine (growth for 3 days). (**C**) Relative growth inhibition of colonies by the same substances. Bars: SD. Statistical significance of differences from the wild type was tested by Student’s *t*-tests (* *p* ≤ 0.05, ** *p* ≤ 0.01, *** *p* ≤ 0.001). Concentrations are expressed in mM, except for iprodione (μg/mL). All experiments were performed in triplicate.

**Figure 5 jof-07-00740-f005:**
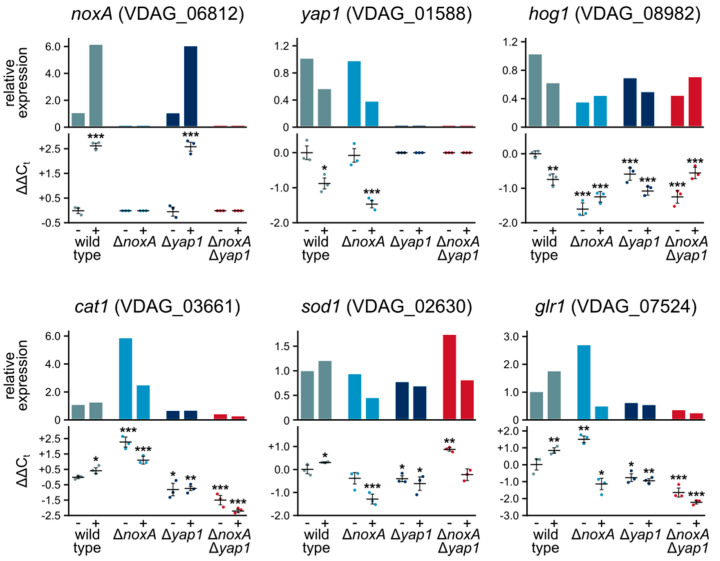
Expression profiles of genes involved in ROS metabolism before (-) and after (+) treatment with 1.5 mM H_2_O_2_. For each gene, the upper plot shows the average relative expression level in each strain compared to the untreated wild-type strain, and the lower shows the determined ΔΔCt values. Three biological replicates (and two technical replicates for each) were performed and analyzed for each strain and gene. In the ΔΔCt plots, the horizontal bars represent mean ΔΔCt values and the bars represent the SD between the three biological replicates. Statistical significance of differences from the untreated wild-type strain was tested by Student’s *t*-tests (* *p* ≤ 0.05, ** *p* ≤ 0.01, *** *p* ≤ 0.001).

**Figure 6 jof-07-00740-f006:**
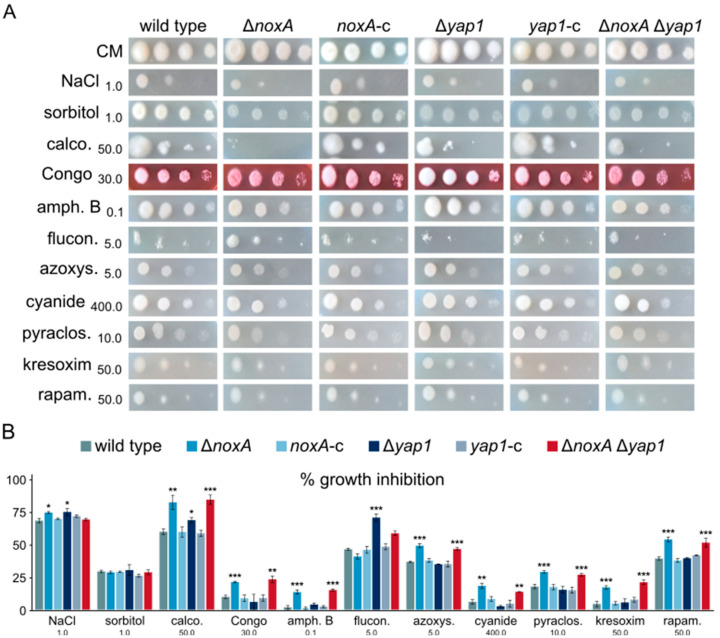
Effects of osmotic stress, cell wall- and plasma membrane-perturbating agents, and antifungal agents that target the respiratory chain on *V. dahliae* Δ*noxA* and Δ*yap1* mutants: (**A**) effects of NaCl, sorbitol, calcofluor white M2R, Congo red, amphotericin B, fluconazole, azoxystrobin, sodium cyanide, pyraclostrobin, kresoxim-methyl, and rapamycin on conidial germination (growth for 3 days). (**B**) Relative growth inhibition of colonies caused by the same substances. Bars: SD. Statistical significance of differences from the wild type was tested by Student’s *t*-tests (* *p* ≤ 0.05, ** *p* ≤ 0.01, *** *p* ≤ 0.001). All concentrations are expressed in μg/mL, except for NaCl (M), sorbitol (M), and rapamycin (nM). All experiments were performed in triplicate.

**Figure 7 jof-07-00740-f007:**
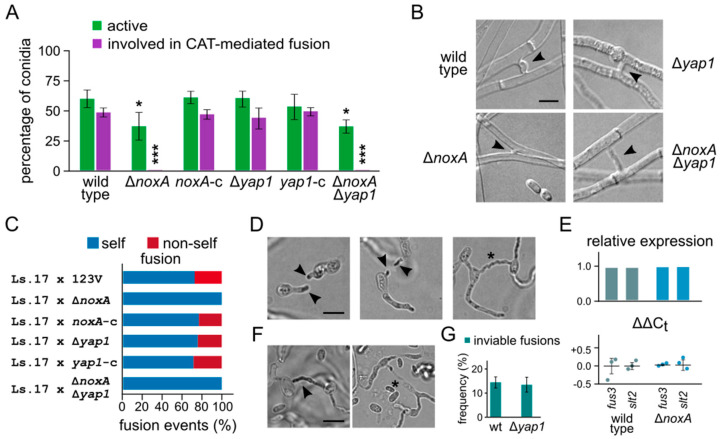
Roles of *V. dahliae* NoxA and Yap1 in somatic cell fusion and the heterokaryon incompatibility reaction: (**A**) competence of deletion mutants in CAT-mediated fusion of conidia or germlings. Frequencies of active conidia and their fraction that are involved in CAT fusion are shown for each strain. All assays were performed in triplicate and 150 conidia were analyzed per replicate. Bars: SD. Statistical significance of differences from the wild type was tested by Student’s *t*-tests (* *p* ≤ 0.05, *** *p* ≤ 0.001). (**B**) Fusions between mature hyphae of the indicated strains. (**C**) Frequencies of self and nonself fusions in pairings of the indicated strains with their incompatible partner, Ls.17 H1-sgfp. Each pairing was tested in triplicate, and 150 fusion events were analyzed per replicate. (**D**) Detection of O_2_^•−^ by NBT staining during CAT homing and establishment of CAT fusion (wild-type strain 123V). Arrowheads: localized ROS production at CAT tips during homing; asterisk: a complete CAT fusion. Bar = 5 μm. (**E**) Relative expression levels of the MAPK genes *fus3* and *slt2* in the wild-type and Δ*noxA* strains. Τhe upper plot shows the average relative expression levels compared to the corresponding genes of the wild type, and the lower shows the determined ΔΔCt values. Three biological replicates (and two technical replicates for each) were analyzed for each strain and gene. In the ΔΔCt plots, the horizontal bars represent mean ΔΔCt values, and the bars SD between the three biological replicates. Statistical significance of differences from the untreated wild-type strain was tested by Student’s *t*-tests (** *p* ≤ 0.01, *** *p* ≤ 0.001). (**F**) Detection of O_2_^•−^ by NBT staining (arrowhead) during the heterokaryon incompatibility reaction (123V × Ls.17 pairing). Inviable cells were no longer stained with NBT (asterisk). (**G**) Frequency of inviable fusions determined by staining with methylene blue in 123V × Ls.17 and 123V Δ*yap1* × Ls.17 pairings. Each pairing was tested in triplicate (n = 100 anastomoses per replicate). Bar = 5 μm.

## Data Availability

The data that support the findings of this study are available in the main text, figures, and additional files of this article.

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
