# Peer review of "The NADPH Oxidase A of Verticillium dahliae Is Essential for Pathogenicity, Normal Development, and Stress Tolerance, and It Interacts with Yap1 to Regulate Redox Homeostasis"

_jof, 2021, doi:10.3390/jof7090740_

Round 1

Reviewer 1 Report

In general terms the topic of the reviewed article is interesting. The  manuscript was prepared with care and its content contains a lot of valuable information. The work does not raise any scientific or substantive reservations.

The manuscript is well structured, the methodology is explicitly presented and the results reported are interesting.

All tables and figures are clear, understandable and necessary. The references are sufficient and necessary.

The paper needs some editorial corrections.

I think that the paper can be published in Journal of Fungi after minor revision.

Author Response

Dear Editor,
Please find attached our revised manuscript “The NADPH Oxidase A of Verticillium dahliae is Essential for Pathogenicity, Normal Development, and Stress Tolerance, and it Interacts with Yap1 to Regulate Redox Homeostasis”, which we would like you to consider for publication in the Journal of Fungi. We would like to express our thanks to the two referees for their time and effort in reviewing our manuscript. Please find our response to their comments below.

Reviewer 1

In general terms the topic of the reviewed article is interesting. The manuscript was prepared with care and its content contains a lot of valuable information. The work does not raise any scientific or substantive reservations.
The manuscript is well structured, the methodology is explicitly presented and the results reported are interesting.
All tables and figures are clear, understandable and necessary. The references are sufficient and necessary.
The paper needs some editorial corrections.
I think that the paper can be published in Journal of Fungi after minor revision.

- We are thankful to the reviewer for the positive feedback.

Reviewer 2 Report

Dear Authors

The manuscript “The NADPH Oxidase A of Verticillium dahliae is Essential for Pathogenicity, Normal Development, and Stress Tolerance, and it Interacts with Yap1 to Regulate Redox Homeostasis” is an interesting and well written article. In this manuscript, authors attempted and performed functional characterization of noxA and its potential role in the biology of V. dahliae.

This manuscript has lot of evidence and supporting data. There remain some minor issues that

authors should consider the comments useful for further revision of the manuscript.

Importantly, there are some grammatical errors throughout the manuscript and so it has

to read by a native speaker for clarity. It is very important to fix all the grammatical

errors before resubmitting the manuscript.

Minor comments:

Authors performed experiments at the gene expression levels. However, it is important to confirm the protein expression using western blot. In particular, in the case of single mutant and double mutants, authors need to compare the protein expression of these mutants with that of wild type

Authors performed statistical analysis using student’s t-test. Since the experiments involve a lot of variables, it is important to use ANOVA test.

This manuscript does not describe the possible mechanism. What is the potential mechanism involved in this biological process and what would be the regulation of the gene expression?

Author Response

The manuscript “The NADPH Oxidase A of Verticillium dahliae is Essential for Pathogenicity, Normal Development, and Stress Tolerance, and it Interacts with Yap1 to Regulate Redox Homeostasis” is an interesting and well written article. In this manuscript, authors attempted and performed functional characterization of noxA and its potential role in the biology of V. dahliae.

This manuscript has lot of evidence and supporting data. There remain some minor issues that authors should consider the comments useful for further revision of the manuscript. Importantly, there are some grammatical errors throughout the manuscript and so it has to read by a native speaker for clarity. It is very important to fix all the grammatical errors before resubmitting the manuscript.

- We would like to thank the reviewer for the favorable report.
- Much to our surprise, though, we read the comment about grammatical errors in our submission. Even though this is new to us, we did perform an extensive grammar and spelling check of the whole manuscript, which nevertheless failed to retrieve such errors or instances of non-standard use of English. We then had it proofread by two colleagues of us, both native speakers of English, and their feedback was similarly very positive. Therefore, we would be grateful if the reviewer were willing to provide us with a list of his/her detected “grammatical errors”, or at least some examples of them.

Minor comments:
Authors performed experiments at the gene expression levels. However, it is important to confirm the protein expression using western blot. In particular, in the case of single mutant and double mutants, authors need to compare the protein expression of these mutants with that of wild type

- In this study, we investigated the possible effects of noxA and/or yap1 deletion on the transcriptional levels of a number of genes, including characteristic examples of the ROS detoxification cell machinery. To address this goal, we performed quantitative RT-qPCR analyses of the respective genes, which revealed a number of significant and interesting transcriptional alterations. While we would generally agree with the reviewer that the study of protein expression is important in the context of many biological investigations, it is unfortunately impossible in this case. We would like to bring to the attention of the reviewer that V. dahliae is a non-model fungus, and very few –if any– specific antibodies have ever been raised against its proteins. Furthermore, our paper deals with several target genes, to which specificity of other available antibodies (i.e. from other species) has never yet been tested. In addition, the alternative of tagging all those genes for performing Western blots with standard antibodies would be prohibitively time-consuming, since genetic engineering of V. dahliae loci remains a considerably laborious endeavour. Furthermore, the latter would also obviously require extensive and time-consuming phenotypic investigation of any tagged strains to exclude undesired function-altering effects of gene tagging. Finally, we would not expect such experiments to add much information regarding our particular research goals in this study.
Authors performed statistical analysis using student’s t-test. Since the experiments involve a lot of variables, it is important to use ANOVA test.
- Although some of our experiments involved multiple variables, most of the biologically meaningful comparisons that we performed (usually between a single group of data and its corresponding control dataset) were pairwise comparisons between two sets of data/variables, while no multiple comparison between different tested groups was meaningful or desired. For the determination of the statistical significance of such comparisons, we used Student’s t-tests, after ensuring that the data satisfied the required conditions for such parametric analyses (and appropriately mentioned this where relevant, i.e. in the legends of Figures 1, 4, 5, 6, 7, and S3). On the other hand, when multiple comparisons were performed, we used the ANOVA test, according to the reviewer’s suggestion (Figure 2).
This manuscript does not describe the possible mechanism. What is the potential mechanism involved in this biological process and what would be the regulation of the gene expression?
- The conserved transcription factor Yap1 is well-established as a major regulator of the antioxidant response in diverse eukaryotic organisms (more information and relevant citations can be found in the Introduction and Discussion of our manuscript). A considerable body of literature has shown that this regulator is involved in the transcriptional control of several antioxidant genes, as well as genes implicated in developmental processes, nutrient utilization, and secondary metabolism. Consistently, we found that
deletion of the V. dahliae yap1 homolog resulted in downregulation of key antioxidant genes. Therefore, the involved mechanism is most likely direct transcriptional regulation, and this is discussed in our manuscript (lines 573-596 in the updated Discussion).
- Regarding the ROS-generating enzyme NoxA, we showed pleiotropic roles in development, stress response, and pathogenicity of V. dahliae. At the core of our hypotheses about the most likely molecular mechanism that mediates these effects lies the fact that ROS can act as signaling molecules involved in the regulation of these processes. The roles of NoxA in the aforementioned biological processes were described in our original submission (lines 525-533, 541-543, 555-560, 568-572, 605-610, 618-622), and more comprehensively discussed in the revised version, according to the reviewer’s suggestion (content added between lines 547-552 of the updated Discussion: “These observations could be likely attributed to the expected function of ROS as secondary signals that could potentially control the expression and function of multiple transcription factors and signaling components via post-translational modification [2]. A possible mechanism for this could involve the control of cellular localization of key signaling components, such as the MAP kinase Mpk1 in P. anserina, whose normal localization depends on nox1 [65].”).